# Application of Cu*_x_*O-Fe*_y_*O*_z_* Nanocatalysts in Ethynylation of Formaldehyde

**DOI:** 10.3390/nano9091301

**Published:** 2019-09-11

**Authors:** Haitao Li, Lijun Ban, Zhuzhu Niu, Xin Huang, Pingfan Meng, Xudong Han, Yin Zhang, Hongxi Zhang, Yongxiang Zhao

**Affiliations:** Engineering Research Center of Ministry of Education for Fine Chemicals, School of Chemistry and Chemical Engineering, Shanxi University, Taiyuan 030006, China; banlijun1992@163.com (L.B.); niuzhuzhu163@163.com (Z.N.); hx06592661@163.com (X.H.); mpf15091823973@163.com (P.M.); hxd5609@163.com (X.H.); sxuzhy@sxu.edu.cn (Y.Z.); zhanghx@163.com (H.Z.)

**Keywords:** Cu*_x_*O-Fe*_y_*O*_z_* complex, formaldehyde ethynylation, core–shell, 2-butyne-1,4-diol

## Abstract

Composite nanomaterials have been widely used in catalysis because of their attractive properties and various functions. Among them, the preparation of composite nanomaterials by redox has attracted much attention. In this work, pure Cu_2_O was prepared by liquid phase reduction with Cu(NO_3_)_2_ as the copper source, NaOH as a precipitator, and sodium ascorbate as the reductant. With Fe(NO_3_)_3_ as the iron source and solid-state phase reaction between Fe^3+^ and Cu_2_O, Cu*_x_*O-Fe*_y_*O*_z_* nanocatalysts with different Fe/Cu ratios were prepared. The effects of the Fe/Cu ratio on the structure of Cu*_x_*O-Fe*_y_*O*_z_* nanocatalysts were studied by means of X-ray diffraction (XRD), Fourier transform infrared spectroscopy (FTIR), ultraviolet confocal Raman (Raman), scanning electron microscopy (SEM), X-ray photoelectron spectroscopy (XPS, XAES), and hydrogen temperature-programmed reduction (H_2_-TPR). Furthermore, the structure–activity relationship between the structure of Cu*_x_*O-Fe*_y_*O*_z_* nanocatalysts and the performance of formaldehyde ethynylation was discussed. The results show that Fe^3+^ deposited preferentially on the edges and corners of the Cu_2_O surface, and a redox reaction between Fe^3+^ and Cu^+^ occurred, forming Cu*_x_*O-Fe*_y_*O*_z_* nanoparticles containing Cu^+^, Cu^2+^, Fe^2+^, and Fe^3+^. With the increase of the Fe/Cu ratio, the content of Cu*_x_*O-Fe*_y_*O*_z_* increased. When the Fe/Cu ratio reached 0.8, a core–shell structure with Cu_2_O inside and a Cu*_x_*O-Fe*_y_*O*_z_* coating on the outside was formed. Because of the large physical surface area and the heterogeneous structure formed by Cu*_x_*O-Fe*_y_*O*_z_*, the formation of nonactive Cu metal is inhibited, and the most active species of Cu^+^ are exposed on the surface, showing the best formaldehyde ethynylation activity.

## 1. Introduction

1,4-butynediol (BYD) is a kind of alkynediol compound with C≡C and OH groups. It has active chemical properties and can be widely used in electroplating solutions, artificial leathers, medicine, pesticides, and other fields [1,2,3,4]. BYD hydrogenation can produce 1,4-butanediol (BDO), 1,4-butenediol, and so on. Hydrogenation products can be further extended downstream to tetrahydrofuran (THF), gamma-butyrolactone (gamma-GBL), polyurethane (PU), polybutylene terephthalate (PBT), polybutylene succinate (PBS), and so on [3,4,5,6]. In recent years, with the rapid development of downstream polyester materials, pharmaceutical intermediates, and other industries, the quality and demand for 1,4-butynediol has increased year by year [7].

1,4-butanediol is synthesized from formaldehyde and acetylene by the Reppe method with copper-based catalysts in industry. The development of high-performance Cu-based ethynylation catalysts has attracted much attention. From the perspective of carrier selection, Cu-based catalysts supported on diatomite, silica gel, activated carbon, and transition or hydrated Al_2_O_3_ and SiO_2_-MgO complexes have been reported in the literature [8,9,10,11,12]. In recent years, we have studied the new CuO-Bi_2_O_3_/SiO_2_-MgO aerogel catalyst, the magnetic separation CuO-Bi_2_O_3_/Fe_3_O_4_-SiO_2_-MgO catalyst, and the core–shell structure CuO-Bi_2_O_3_@SiO_2_ catalyst. The catalyst supported on TiO_2_ showed better activity and stability in the formaldehyde ethynylation reaction [13,14,15,16,17,18].

Another widely used catalyst for formaldehyde ethynylation is nano-Cu-based powder catalysts without support (CuO or Cu_2_O). In a slurry bed reactor, nano-Cu-based powder catalysts often show higher performances. It was found that the crystallinity of Cu_2_O species could be well regulated by controlling the precipitation conditions and methods, and then Cu_2_O could be efficiently converted into an active component, cuprous acetylide, in the formaldehyde ethynylation reaction, thus significantly improving the catalytic activity of the catalyst. Later research confirmed that the introduction of additives into CuO to prepare multicomponent composite nanocatalysts can not only improve the dispersion of copper species, but the additives can further improve the catalytic activity through electronic effects or synergistic adsorption. The introduction of Bi_2_O_3_ can effectively inhibit the excessive reduction of copper species to metal Cu^0^ in the formaldehyde ethynylation reaction, which results in catalyst deactivation [17]. The introduction of MgO enhances the surface alkalinity of the catalyst. The synergistic effect of a Cu center and a base center promotes the activation and adsorption of acetylene, and it makes the catalyst show higher catalytic activity [19]. The activation and adsorption of formaldehyde molecules has been realized by introducing copper centers and zinc oxide. The ethynylation performance of formaldehyde has further improved [20].

As an n-type semiconductor, Fe_2_O_3_ has been widely used in the photocatalytic field. It has been found that introducing Fe_2_O_3_ into Cu_2_O can form p–n heterostructures [21], which have strong interactions and transfer electrons between Cu species and Fe species, and helps to regulate the reducibility of Cu species. In the magnetically separated CuO-Bi_2_O_3_/Fe_3_O_4_-SiO_2_-MgO catalysts prepared by Wang Junjun et al. [16], the introduction of Fe_3_O_4_ not only played a role in magnetic separation, but also affected the existence of Cu species through the effect of electronic promoters, which led to an excellent formaldehyde ethynylation performance. Fe is considered as a good additive for formaldehyde ethynylation. Fe^2+^ and Fe^3+^ are reducible and oxidative in Fe_3_O_4_, while Cu^2+^ is converted to Cu^+^ under the reductive condition of formaldehyde ethynylation, and the formation of inactive over-reductive metal Cu products should be inhibited. Cu*_x_*O-Fe*_y_*O*_z_* nanocatalysts containing Cu^+^, Cu^2+^, Fe^2+^, and Fe^3+^ were prepared. The stability of the active sites of Cu^+^ in a reducing atmosphere could be improved by using the electronic effects among species with different valences; thus, the activity and stability of the formaldehyde ethynylation reaction could be improved.

A variety of synthetic protocols in both aqueous and nonaqueous media have been developed for large-scale synthesis of composite nanostructure catalysts. Redox transformation reactions have shown great potential for the synthesis of hollow/porous metal oxide nanoframeworks, alloy nanostructures, and composite nanostructure catalysts [22,23,24,25,26]. Recently, Pal et al. [27] reported a systematic and delicate size- and shape-controlled synthesis of CuO-MnO_2_ composite nanostructure catalysts from time-dependent redox transformation reactions between Cu_2_O and KMnO_4_. The composite nanostructure catalysts acted as efficient recyclable catalysts for nitroarene reduction in water at room temperature. Fe^3+^ is an oxidizing metal similar to KMnO_4_. In this work, a series of Cu*_x_*O-Fe*_y_*O*_z_* catalysts with different Fe/Cu ratios were prepared by the redox reaction between Cu_2_O and Fe^3+^ using iron nitrate as the Fe^3+^ source. The formation mechanism of the catalysts and the effects of the Fe/Cu ratio on the structure, texture, valence state of Cu species, reduction performance, and ethynylation activity of the catalysts were studied.

## 2. Materials and Methods

### 2.1. Preperation of Catalysts

All chemical reagents, Cu(NO_3_)_2_·3H_2_O(Macklin), Fe(NO_3_)_3_·9H_2_O(Macklin), PEG-600(Macklin) and NaOH(Macklin), were of analytic grade and used as-purchased without pre-purification.

Cu_2_O: 100 mL of 0.125 mol/L Cu(NO_3_)_2_·3H_2_O solution was transferred into a 1000 mL beaker, and the beaker was placed in a constant temperature water bath at 30 °C. PEG-600 (100 mL) was added to the above solution under magnetic stirring. A total of 150 mL of 1.67 mol/L NaOH solution was measured and poured into the beaker. After magnetic stirring for 5 min, 300 mL of 0.25 mol/L sodium L-ascorbate solution was added dropwise, and the dropping rate was 1.25 mL/min. Then, the mixture was stirred continually for 30 min and was allowed to stand for 1 h at a constant temperature. After being centrifuged, the mixture was washed three times with distilled water and absolute ethanol, and it was dried in a vacuum dryer at 60 °C for 4 h. The obtained sample was marked as Cu_2_O.

Cu*_x_*O-Fe*_y_*O*_z_*: 2 g of the above prepared Cu_2_O powder was dispersed in 10 mL of Fe(NO_3_)_3_·9H_2_O aqueous solution. The solution was sonicated at room temperature for 20 min, magnetically stirred for 30 min, and centrifuged to obtain the solid material. The solid material was dried at 60 °C for 10 h in a vacuum dryer, and it was calcined at 300 °C for 3 h under a nitrogen atmosphere to form Cu*_x_*O-Fe*_y_*O*_z_*. The concentrations of Fe(NO_3_)_3_·9H_2_O aqueous solution in the above preparation procedure were adjusted to 0.01, 0.1, 1, and 2 mol/L, respectively, and a series of different Fe/Cu ratios of Cu*_x_*O-Fe*_y_*O*_z_* were synthesized. The Fe/Cu molar ratios of the five catalysts prepared at different concentrations were determined by inductively coupled plasma atomic emission spectroscopy (ICP) to be 0.05, 0.1, 0.8, and 2, respectively, and the samples were marked as CF-0.05, CF-0.1, CF-0.8, and CF-2.

### 2.2. Characterization of Catalysts

N_2_-Physisorption analyses were performed using a Micromeritics ASAP-2020 apparatus (Norcross, GA, USA). The fresh and used samples were degassed at 150 °C for 5 h and 60 °C for 24 h, respectively, prior to adsorption testing. Specific surface areas were obtained using the multipoint Barrett-Emmett-Teller (BET) procedure.

X-ray diffraction patterns of the samples were recorded with a D8 Advance diffractometer (Bruker Corporation, Billerica, MA, USA) with Cu Kα radiation (λ = 1.5418 Å).

Fourier transform infrared (FTIR) spectra were recorded with a Nicolet™ iS50 spectrophotometer (Thermo Fisher Scientific, Waltham, MA, USA) in the range of 400–4000 cm^−1^.

Scanning electron microscopy (SEM) images were obtained with Hitachi S4800 electron microscope operated at beam energy of 3.0 kV.

Raman spectroscopy was performed with a LabRAM HR Evolution Raman spectrograph (HORIBA Scientific, Paris, France) with a 532 nm laser operated at 0.08 mW.

X-ray photoelectron spectroscopic (XPS) measurements were conducted on an ESCALAB 250 spectrometer (Thermo Fisher Scientific, Waltham, MA, USA) using an Al Kα X-ray source (hν = 1486.7 eV).

Auger electron spectroscopy (XAES) was performed on a PHI 1600 ESCA spectrometer (Perkin-Elmer, Waltham, MA, USA) equipped with a monochromatic Al Ka X-ray source (hm = 1361 eV) operating at a pass energy of 100 eV.

H_2_ temperature-programmed reduction (H_2_-TPR) experiments were performed on a Micromeritics AutoChem II 2920 automatic temperature-programmed chemical adsorption instrument (Norcross, GA, USA), and about 30 mg of the sample was loaded into a quartz U-tube for each measurement. Prior to the measurement, the sample was treated at 350 °C for 2 h under Ar stream. When the temperature was dropped to 50 °C, the H_2_-Ar mixture (5% H_2_ by volume) was switched on, and the temperature increased with a ramp of 10 °C/min.

### 2.3. Catalyst Test of Catalysts

The catalyst was evaluated in a three-neck flask connected with a reflux condenser. A blend of a quantity of catalyst with 50 mL of formaldehyde (35 wt.%) aqueous solution was carried out in an oil-bath flask under electromagnetic stirring. An adequate quantity of pure N_2_ was poured into the flask to purge O_2_, and the solution containing the catalyst was heated to reaction temperature (90 °C) under continuous agitation. A C_2_H_2_ stream was then turned on to initiate the ethynylation reaction. After several hours, by lowering the reaction temperature to room temperature, the catalytic reaction was stopped, an N_2_ stream was poured, and the C_2_H_2_ stream was shut down.

The used catalyst was centrifuged, washed in deionized water, and dried in a vacuum at ambient temperature. In the centrifugal solution, the formaldehyde content was titrated with sodium thiosulfite to get the conversion rate of formaldehyde. The analysis of 1,4-butynediol content on an Agilent 7890A gas chromatography (Santa Clara, CA, USA) was conducted using the internal standard method with the addition of 1,4-butanediol. The selectivity of 1,4-butynediol was obtained through the method of dividing the yield of 1,4-butynediol by the conversion rate of formaldehyde [7].

## 3. Results

### 3.1. Structure Analysis of Catalysts

The phase and composition of catalysts were characterized by X-ray diffractometry (XRD), and the obtained XRD patterns are shown in Figure 1. Peaks obtained in the XRD pattern confirmed the crystalline formation of Cu_2_O. The sharp characteristic diffraction peaks at the 2θ angles of 29.6, 36.5, 42.4, 61.4, 73.6, and 77.4 can be indexed to (110), (111), (200), (220), (311), and (222) planes, respectively, of the cubic cuprous oxide phase (JCPDS card No. 05-0667) [28]. After loading of Fe during the reaction process, the intensity of the Cu_2_O peaks became gradually weaker with the increase of the Fe/Cu weight ratio. The Cu_2_O peaks in CF-2 almost disappeared. The diffused characteristic diffraction peaks located at the 2θ angles 35.5 and 38.7 can be assigned to the diffraction of Fe_2_O_3_ nanoparticles, indicating that the Cu_2_O content decreased, and the dispersion increased. The loaded Fe_2_O_3_ existed in an amorphous or highly dispersed form. It is speculated that the loaded Fe species was not simply adsorbed on the Cu_2_O surface, but it reacted with Cu_2_O, causing a change in the state of Cu_2_O itself.

### 3.2. FT-IR Analysis and Raman Analysis

Figure 2 represents the FT-IR spectrum of the Cu_2_O and Cu*_x_*O-Fe*_y_*O*_z_* catalysts with different Fe/Cu weight ratios. As shown in the infrared spectrum of Cu_2_O, the infrared absorption peak observed at 622 cm^−1^ was assigned to Cu(I)–O bond stretching and bending vibration modes [29]. After loading of Fe, the absorption peaks centered at 475 cm^−1^ and 552 cm^−1^ were assigned to the Fe^3+^–O^2−^ bond stretching in the FeO_6_ octahedron and Fe^3+^–O^2−^ bond stretching in the FeO_4_ tetrahedron, respectively [30]. The infrared absorption peak of Cu_2_O observed at 622 cm^−1^ became weaker with the increase of Fe/Cu weight ratio; the peak almost disappeared when the Fe/Cu weight ratio was 2, indicating the decrease of relative content of Cu_2_O or the conversion of species.

Raman spectroscopy has been widely used to characterize crystal structures and for qualitative analysis. Cu_2_O has a simple cubic lattice and belongs to the O_4h_ space group [31]. According to the space group theory, Cu_2_O has six vibration modes, Γ = F_2g_ + 2F_1u_ + F_2u_ + E_u_ + A_2u_. Theoretically, for a perfect Cu_2_O crystal, only the F_2g_ vibration mode has Raman activity, and the two F_1u_ vibration modes have infrared activity. However, due to the lattice defects, not only the intensity of the intrinsic peaks is reduced or even masked, but also the vibrational modes of non-Raman activity are excited. These have become an important basis for identifying Cu_2_O species [32,33]. Figure 3 is a Raman spectrum of the catalyst. The characteristic Raman shifts of the Cu_2_O catalyst can be observed at 148, 218, 414, 520, and 631 cm^−1^. According to the literature, the observed Raman peaks can be assigned to the F2g vibration mode at 520 cm^−1^, the infrared active F1u vibration modes at 148 and 631 cm^−1^, and the multiphonon vibration modes at 218 and 414 cm^−1^. With the increase of the Fe/Cu weight ratio, the intensity of the Raman vibration peak of Cu_2_O at 218 cm^−1^ decreased, and it even disappeared when Fe/Cu was 0.8. At the same time, the diffused Raman shift assigned to the Fe_2_O_3_ species can be observed at 248 and 720 cm^−1^ [34,35]. This indicates that the initial loaded low content of Fe led to a decrease in the crystallinity of the Cu_2_O surface, and the loading of a large amount of Fe_2_O_3_ made the surface of Cu_2_O highly dispersed or covered by Fe_2_O_3_. Comparing the XRD spectrum with the Raman spectrum of the CF-0.8 sample whose Fe/Cu ratio was 0.8, it can be found that a clear crystal diffraction peak assigned to Cu_2_O was still observed in XRD, and in the Raman spectrum, the characteristic Raman shift assigned to Cu_2_O disappeared. Considering that the XRD spectrum depicts phase characteristics, while the Raman spectra gives the characteristics of the surface information of the sample, it can be speculated that CF-0.8 formed the amorphous Cu-Fe composite at the surface, and the interior was still a covering structure of the Cu_2_O crystal. Comparing the XRD with Raman characterization data of each sample with the increase of the Fe/Cu ratio, it is further speculated that there was a reaction between Fe^3+^ and Cu^+^, and the reaction gradually expanded from the surface of Cu_2_O to the bulk phase until complete conversion.

### 3.3. Morphological Analysis

To observe whether Fe-doping had an impact on the surface structure, crystal phase, and chemical composition of the catalyst, the catalyst was characterized by SEM, and an EDS element analysis was carried out on the selection. The results are shown in Figure 4 and Table 1. Figure 4a is an SEM image of Cu_2_O. It can be observed that the pure Cu_2_O crystal had an octahedral morphology [36], and the particles were intact and uniform in size. According to the EDS analyses of the selected crystal plane a1 and the vertex corner a2, it was discovered that only the Cu element exists (Table 1). After Fe was introduced into Cu_2_O, and when the Fe/Cu ratio was 0.05, it was found that irregular particles were attached at the edges, arrises, and corners of the Cu_2_O crystal, and the Cu_2_O crystal plane maintained the original features. According to the respective EDS analyses of the selected newly formed, irregular particles b1 and Cu_2_O crystal plane b2, it was shown that only Cu existed in the b2 area, and both Fe and Cu elements existed in the b1 area. When the Fe/Cu ratio reached 0.1, the irregular particles in the CF-0.1 sample increased, while the Cu_2_O crystal plane decreased. The proportion of irregular particles increased from 13.8% to 21.9% with an Fe/Cu ratio of 0.05. This result indicated that when Fe^3+^ was introduced into Cu_2_O, the irregular particles formed were not the physical adhesion of the formed Fe_2_O_3_, but the nanoparticles were formed by the reaction between Cu_2_O and Fe^3+^. Since the edges, arrises, and corners of the Cu_2_O crystal are at a high energy, the composite nanoparticles are preferentially formed here, and as the Fe/Cu ratio increased, the exposed Cu_2_O crystal plane decreased. The exposed Cu_2_O crystal plane still existed in the SEM image of the CF-0.1 sample, which corresponds to the obvious Cu_2_O characteristic peak in the XRD and Raman spectra. When the Fe/Cu ratio was increased to 0.8, it was discovered in the CF-0.8 sample that the surface of the Cu_2_O crystal was completely destroyed, and the overall surface showed a rough spherical morphology. After EDS tests, the surface chemical composition was evenly distributed, and the Fe/Cu ratio in different areas was 1 (or so). Combined with XRD and Raman characterizations, the XRD spectrum of Cu_2_O could still be observed in the CF-0.8 sample, while the Raman shifts disappeared, indicating that the core–shell structure with an inner core of Cu_2_O and an outer shell of Fe-Cu composite formed. By further increasing the Fe/Cu ratio, a rough, spherical morphology can be observed in the CF-2 sample, and Cu_2_O reacted completely with the introduced Fe to form a phase Fe-Cu composite.

### 3.4. Surface Composition and Valence Analysis of the Catalyst

XPS characterization is widely used in catalyst material analysis. To determine the composition of samples with different Fe/Cu ratios, XPS usually has a certain degree of transparency to the surface, and the Cu-O system can reach 9–10 monolayers.

The Cu2p images of Cu_2_O, CF-0.8, and CF-2 were selected for peak fitting, and the peak positions and surface atomic ratios of different valence elements were calculated according to the fitting images, which are shown in Figure 5 and Table 2. It can be seen from the data of the peak fitting that Cu_2_O, CF-0.8, and CF-2 had two XPS characteristic peaks at 932.5 and 952.4 eV, which correspond to the binding energies of Cu2p_3/2_ and Cu2p_1/2_ in Cu_2_O. Two XPS characteristic peaks also appeared at higher binding energies (934.5 and 954.6 eV), corresponding to Cu 2p_3/2_ and Cu 2p_1/2_ in CuO, respectively [37,38]. In addition, the 2p → 3d satellite peak of Cu^2+^ appeared at 941–945 eV, further confirming the presence of CuO species on the surface of the sample. According to the corresponding integral data of the Cu^2+^→Cu^+^ peak area at Cu2p_3/2_, the atomic ratios of Cu^2+^/Cu^+^ on different samples were calculated (Table 2). It can be seen that there was also Cu^2+^ in pure CuO, and the ratio of Cu^2+^/Cu^+^ was 0.31, which may be due to the oxidation of partial Cu_2_O surface by O_2_ to CuO during sample preparation. With the increase of Fe content, the ratio of Cu^2+^/Cu^+^ on the surface of the catalyst increased, and the ratio of Cu^2+^/Cu^+^ in CF-2 reached 2.3, indicating that more CuO formed. At the same time, it was found that in CF-0.8 and CF-2 prepared by introducing Fe, the binding energy at Cu2p_3/2_ corresponding to Cu^+^ shifted to a high binding energy of 932.8 eV.

According to the peak fitting for the Fe2p XPS images of CF-0.05, CF-0.8, and CF-2 catalysts, it can be seen that all catalysts had two XPS characteristic peaks at 711.6 and 725.5 eV, corresponding to the binding energies of Fe^3+^2p_3/2_ and Fe^3+^2p_1/2_ in Fe_2_O_3_ [39], indicating that most of the Fe species in the prepared catalysts existed in the form of Fe^3+^. Two XPS characteristic peaks also appeared at 710.1 and 724 eV, corresponding to the binding energies of Fe^2+^2p_3/2_ and Fe^2+^2p_1/2_ [40]. The Fe^2+^/Fe^3+^ atomic ratio data calculated from the peak areas of Fe^2+^ and Fe^3+^ at the Fe2p_3/2_ peak are listed in Table 2. It can be seen that the surface Fe^2+^/Fe^3+^ ratio increased with the increase of the Fe/Cu ratio because more Fe^3+^ was converted to Fe^2+^ in the reaction process. In Appendix A, it is shown that as the Fe/Cu ratio was lower than 0.8, the peak area of Fe2p increased with the increase of the Fe/Cu ratio, which was attributed to the enrichment of Fe on the surface of the catalyst particles. When the Fe/Cu ratio was further increased by 2, the peak area of Fe2p decreased. In response to this phenomenon, we analyzed the O/Fe+Cu+O ratio on each sample, and the results are listed in supporting information (Appendix A). Appendix A shows that the ratio of O/Fe+Cu+O increased with the increase of the Fe/Cu ratio. We speculate that the Fe2p integrated area of CF-2 decreased because of the increased O content. The increase of O content in the surface was due to the conversion of Cu_2_O to CuO and the increase of O atoms bound to Fe^3+^. Synthesized with the results of the increase of Cu^2+^/Cu^+^ and Fe^2+^/Fe^3+^ ratios in the sample introduced into Fe, it is speculated that a redox reaction occurred between Cu_2_O and Fe^3+^, and a Cu*_x_*O-Fe*_y_*O*_z_* shell formed on the outer surface of Cu_2_O. In addition, after the introduction of Fe, the Cu^+^ electron binding energy increased. It was also indicated that a heterostructure formed between Cu*_x_*O-Fe*_y_*O*_z_*, which decreased the density of the Cu^+^ electron cloud and increased the binding energy of the Cu^+^ electron cloud.

### 3.5. Discussion on the Formation Mechanism of Cu_x_O-Fe_y_O_z_ Structure

By virtue of the redox reaction between Cu_2_O and Fe^3+^, Cu*_x_*O-Fe*_y_*O*_z_* can be prepared by controlling the concentration of the initial Fe^3+^. The synthesis and morphology control of Cu_2_O has been fully discussed in the literature and is not discussed in this work [41,42,43,44]. When the concentration of the initial Fe(NO_3_)_3_·9H_2_O solution was 0.8 mol/L, the prepared catalyst exhibited a core–shell structure, where the inner layer was Cu_2_O and the outer layer was coated with Cu*_x_*O-Fe*_y_*O*_z_* (Figure 4d). Based on the XPS analysis of the sample, the outer structure contained two metal elements, Fe and Cu. Fe presents two valence states of Fe^2+^ and Fe^3+^, while Cu exhibits valence states of Cu^2+^ and Cu^+^ (Figure 5). In the initially prepared Cu_2_O, there were also CuO species derived from the oxidation of O_2_ during the preparation procedures, and the CuO species increased significantly after Fe^3+^ treatment. At the same time, under the condition of nonisolating O_2_, the reduced state product Fe^2+^ of Fe^3+^ was produced, which was inevitably derived from the reduction of Cu^+^. The standard reduction potential of Fe^3+^/Fe^2+^ was 0.771V, while the standard reduction potential of Cu^2+^/Cu^+^ was +0.203 V. From the perspective of thermodynamics, the processes of oxidizing Cu^+^ to Cu^2+^ by Fe^3+^ and converting Fe^3+^ itself into Fe^2+^ can be spontaneously carried out:Fe^3+^ + Cu^+^ → Fe^2+^ + Cu^2+^.(1)

In the presence of water, it can be written in steps:Fe^3+^ + 2OH^−^ + e^−^→ Fe(OH)_2_(s) →FeO + H_2_O;(2)
Cu_2_O + 2H_2_O + 1/2O_2_ → 2Cu(OH)_2_(s) → 2CuO + 2H_2_O.(3)

Cu^+^ is oxidized to Cu^2+^, and Fe^3+^ is converted to Fe^2+^ under the condition of nonisolating O_2_, but these redox reactions selected in this work cannot be completely carried out. In the outer surface composition of the samples with an Fe/Cu ratio of 0.8 and Fe/Cu ratio of 2, Cu^+^, Cu^2+^, Fe^3+^, and Fe^2+^ exist simultaneously, indicating that a redox balance was reached between them. And under such balance, it is difficult to further reduce Cu^+^ and Cu^2+^ to a lower valence state and to oxidize Fe^2+^ to a higher valence state. Under the atmosphere of oxidizing O_2_, oxides of different valence states can stably exist, and it is speculated that this composition exhibits good stability under a reducing atmosphere. By comparing the SEM results of Cu*_x_*O-Fe*_y_*O*_z_* prepared by different concentrations of Fe(NO_3_)_3_ solution, it was also found that the redox reaction between Cu_2_O and Fe^3+^ went through the reaction process from the edge/arris/corner to the exposed surface and, finally, to the surface phase. The driving force for nucleation results from the difference of surface free energy [45,46]. Exposed high refractive index surfaces, defects, atomic steps, and kinks can be used as nucleation sites. The edge and vertex of Cu_2_O have higher surface free energy than those of the crystal mask, so it can be preferentially used as nucleation site [47]. At low Fe^3+^ concentrations, Fe^3+^ preferentially reacts with Cu^+^ at the edge, arris, and corner of the Cu_2_O crystal to form amorphous Cu*_x_*O-Fe*_y_*O*_z_* particles, which partially cover only the position of the edge, arris, and corner of Cu_2_O. In the regular Cu_2_O crystal, the edge, arris, and corner are often distributed with more defects and are at higher energy levels, which are the main reason for the preferential oxidation-reduction between Cu_2_O and Fe^3+^.

With the increase of Fe^3+^ concentration, Cu_2_O on the exposed crystal surface reacted with Fe^3+^. When the concentration of Fe^3+^ was 0.8, Cu_2_O converted into an amorphous sphere with a rough surface. XRD showed that its phase structure was still Cu_2_O with complete crystal form, namely, forming the core–shell structure of Cu*_x_*O-Fe*_y_*O*_z_*. When the Fe^3+^ concentration continued to increase, the excess Fe^3+^ concentration migrated to the phase through the amorphous shell and reacted with the phase Cu_2_O to form a new Cu*_x_*O-Fe*_y_*O*_z_* complex, until the Cu_2_O was destroyed as a whole, forming an amorphous Cu*_x_*O-Fe*_y_*O*_z_* and also causing changes in the texture–structural parameters of the catalyst. It can be seen from Appendix A that as the ratio of Fe/Cu increased, the specific surface area and pore volume first increased and then decreased (Figure 6).

### 3.6. Analysis of Reduction Behavior of the Catalyst

The redox performance of the catalyst is one of the important factors affecting the activity of the catalyst. Cu*_x_*O-Fe*_y_*O*_z_* catalysts with different Fe/Cu ratios were characterized by TPR. As can be seen from the Figure 7, the Cu_2_O catalyst had a broad reduction peak in the range of 200 to 300 °C, and the peak temperature was 285 °C. After doping Fe, it was found that when the Fe content was less than 0.8, the reduction peak shapes of each catalyst were similar to that of the Cu_2_O catalyst, and the peak temperature moved to a high-temperature range with the increase of Fe/Cu ratio. The peak temperatures in CF-0.05, CF-0.1, and CF-0.8 were 297, 305, and 325 °C, respectively. These indicate that the doping of Fe did not destroy the crystal phase structure of Cu_2_O, and the reduction peak was the hydrogen-consuming peak of Cu metal reduced from Cu_2_O. The start/stop reduction temperature of the catalyst shifted to the direction of high temperature, revealing that there was a certain interaction in the catalyst, and the corresponding copper-iron heterostructure formed to reduce the reduction of Cu_2_O. Combined with XPS analysis, the peak area of Fe2p increased with the increase of the Fe/Cu ratio (Appendix A). When CF-0.8 reached the maximum, it was shown that the enrichment of Fe on the surface of CF-0.8 reached the maximum, and a strong interaction between copper-iron complexes formed, which led to the highest reduction temperature. Compared with the Cu_2_O catalyst, when the Fe/Cu ratio reached 2, the reduction peak had a large shift to the direction of low temperature. The peak temperature was 238 °C, and the peak shape was irregular. It is speculated from the results of XPS analysis that, perhaps because of the increase of Fe content, the crystal structure of Cu_2_O was destroyed to form the isolated Cu^2+^ and Fe-Cu^+^ and make the reduction peak shift to the direction of low temperature.

### 3.7. Evaluation Results of Catalyst Activity

The reaction of formaldehyde with acetylene can produce 2-propyn-1-ol and 2-butyne-1,4-diol: HC≡CH + HCHO → HC≡CCH_2_-OH → HO-CH_2_C≡CCH_2_-OH. Theoretically, the solubility of C_2_H_2_ gas in HCHO solution is relatively low. In the aqueous phase reaction system, the concentration of HCHO is much larger than that of C_2_H_2_. In the reaction, it is easier for one molecule of acetylene to react with two molecules of formaldehyde to obtain BYD. The formation of propynyl alcohol is often required to be carried out at a high acetylene pressure and in a benign solvent such as γ-butyrolactone (GBL) and N-methylpyrrolidone (NMP). The system with acetylene gas in the atmospheric pressure and aqueous formaldehyde solution used in this work only produced BYD, and no formation of propynyl alcohol was observed. The yield of BYD in the reaction system directly reflects the activity of the catalyst. Figure 8a shows the BYD yield of Cu_2_O and Cu*_x_*O-Fe*_y_*O*_z_* with different ratios. It can be seen that the yield of 2-butyne-1,4-diol in pure Cu_2_O sample was only 21.8%. With the doping of Fe, the yield of 2-butyne-1,4-diol first increased and then decreased. The yield of 2-butyne-1,4-diol in the CF-0.8 sample reached a maximum of 59.83%, and CF-0.8 showed the best catalytic performance. The stability of two catalysts, Cu_2_O and CF-0.8, was investigated. The results are shown in Figure 8b. It can be seen that the yield of 2-butyne-1,4-diol decreased in both catalysts during the 6-cycle test. But the decrease trends were significantly different: the yield of 2-butyne-1,4-diol on the Cu_2_O catalyst decreased from about 22% to about 10%, while the yield of 2-butyne-1,4-diol in the CF-0.8 catalyst decreased from about 60% to about 50%. And in the second cycle test of the Cu_2_O catalyst, the yield of 2-butyne-1,4-diol decreased significantly, while in the CF-0.8 catalyst, the yield of 2-butyne-1,4-diol decreased in the fourth cycle test, indicating that the CF-0.8 catalyst had a higher stability than the Cu_2_O catalyst. By comparing with different catalysts (Appendix A), it was found that the doping of Fe improved the catalytic activity, and the stability of the catalyst, which shows the advantages of Cu*_x_*O-Fe*_y_*O*_z_*.

In ethynylation of formaldehyde, the initial Cu species (Cu^2+^ or Cu^+^) in situ can be converted into the active species of the cuprous acetylide to exhibit catalytic activity. The Cu^+^ site exposed on the surface of the catalyst after the reaction was positively correlated with the ethynylation activity of formaldehyde (i.e., BYD yield). In the Cu*_x_*O-Fe*_y_*O*_z_* catalyst system, exposure of the Cu^+^ site on the surface of the catalyst after activation was determined by the following factors: (1) physical specific surface area of the catalyst, (2) Fe/Cu ratio on the surface of the catalyst, and (3) whether Cu^2+^ or Cu^+^ was excessively reduced to inactive species of metal Cu. Combined with catalyst characterization, it was found that with the introduction of Fe species, the catalyst changed from a regular crystal structure to a coarse, amorphous structure. During this process, the specific surface area and pore volume increased significantly. The large specific surface area is favorable for the exposure of active species, the large pore structure is favorable for the diffusion of reactants and products, and it is advantageous for the improvement of catalytic activity. An unfavorable factor accompanying the change in specific surface is that Fe occupies part of the Cu site, which is not conducive to the exposure of Cu species, and the two factors mutually restrict each other. The specific surface areas and Fe/Cu ratios of the different samples are listed in Appendix A. It can be seen that the specific surface area of Cu_2_O was only 7.7 m^2^/g. After the introduction of Fe, the specific surface area of the sample notably increased. The specific surface area of the CF-0.8 sample reached 13.9 m^2^/g, about twice that of Cu_2_O. At this time, the surface Fe/Cu ratio was 0.79, and the amount of Cu exposed on the surface increased as a whole. When the Fe/Cu ratio further increased, the CF-2 specific surface area further increased to 17.6 m^2^/g. However, it was difficult to compensate for the decrease of surface Cu species because of the increase of the Fe/Cu ratio. This is an important reason for the first increase and then decrease of BYD yield with an increasing Fe/Cu ratio.

On the other hand, it is worth noting that the above-discussed Cu species on the specific surface area contain various Cu species, including the Cu+ species in the active cuprous acetylide converted by ethynylation of formaldehyde and the inactive metal Cu species produced by over-reduction. The inactive metal Cu not only covered the effective active Cu^+^ species but also decreased the active Cu^+^ species. Metallic Cu can catalyze the polymerization of acetylene, resulting in deactivation of carbon deposits on the catalyst surface. Under the premise of increasing the surface Cu species, regulating the chemical environment and reducibility of Cu species and preventing the production of over-reduced species Cu are the important factors to improve the performance of the ethynylation of formaldehyde for the catalyst. In the catalyst of Cu*_x_*O-Fe*_y_*O*_z_* prepared by introducing Fe into Cu_2_O, by virtue of the heterostructure formed between Cu*_x_*O and Fe*_y_*O*_z_*, the electron cloud density of the Cu species was redistributed, a good stability in the redox atmosphere was maintained, and the Cu^+^ species in the ethynylation of formaldehyde existed stably (Figure 5) to restrict the excessive reduction of Cu^+^ to metallic Cu, which is another important factor that the CF-0.8 sample containing Fe had the highest ethynylation activity of formaldehyde. Figure 6 TPR confirmed that the introduction of a proper amount of Fe could regulate the reduction performance of Cu^+^. In the sample after reaction, the XRD peak of metallic Cu appeared in Cu_2_O, and the peak intensity of metallic Cu decreased with the increase of Fe/Cu ratio (Appendix A). The Auger electron spectroscopy after the reaction also showed that the peak area of the metal Cu in the Fe-containing sample decreased significantly (Appendix A). These results confirm the above inference.

## 4. Conclusions

A series of Cu*_x_*O-Fe*_y_*O*_z_* nanocatalysts with different Fe/Cu ratios were prepared by the oxidation–reduction reaction between Cu^2+^ and Fe^3+^ by and using Cu_2_O and Fe(NO_3_)_3_ as raw materials. The introduced Fe^3+^ reacted with Cu_2_O to form Cu^2+^ and transformed itself into Fe^2+^ so that Cu*_x_*O-Fe*_y_*O*_z_* nanocomposite particles preferentially formed at the edges, arrises, and corners of the Cu_2_O crystal. As the Fe/Cu ratio increased, the content of Cu*_x_*O-Fe*_y_*O*_z_* increased, and Cu*_x_*O-Fe*_y_*O*_z_* gradually spread to the exposed crystal surface of Cu_2_O. Moreover, Cu^2+^/Cu^+^, Fe^2+^/Fe^3+^, and Fe/Cu ratios increased in Cu*_x_*O-Fe*_y_*O*_z_*. When the Fe/Cu ratio was 0.8, a core–shell structure of the Cu_2_O@Cu*_x_*O-Fe*_y_*O*_z_* catalyst formed, with an inner core of Cu_2_O and an outer layer of Cu*_x_*O-Fe*_y_*O*_z_*. When the Fe/Cu ratio continued to be increased, the crystal structure of Cu_2_O disappeared, forming a Cu*_x_*O-Fe*_y_*O*_z_* nanocomposite catalyst that was internally and externally uniform. In the catalyst with a Fe/Cu ratio of 0.8, a heterogeneous structure formed between Cu*_x_*O and Fe*_y_*O*_z_*, and the coexistence of Cu^2+^, Cu^+^, Fe^2+^, and Fe^3+^ maintained the stability of Cu^+^ in a reducing atmosphere of formaldehyde ethynylation and inhibited the excess reduction of Cu^+^ to metal Cu species. Meanwhile, the Cu*_x_*O-Fe*_y_*O*_z_* shell with the rough, amorphous structure exhibited a high physical specific surface. Under the combined action of both sides, the Cu^+^ active site exposed to the outer surface was the most prominent and showed the best ethynylation activity.

## Figures and Tables

**Figure 1 nanomaterials-09-01301-f001:**
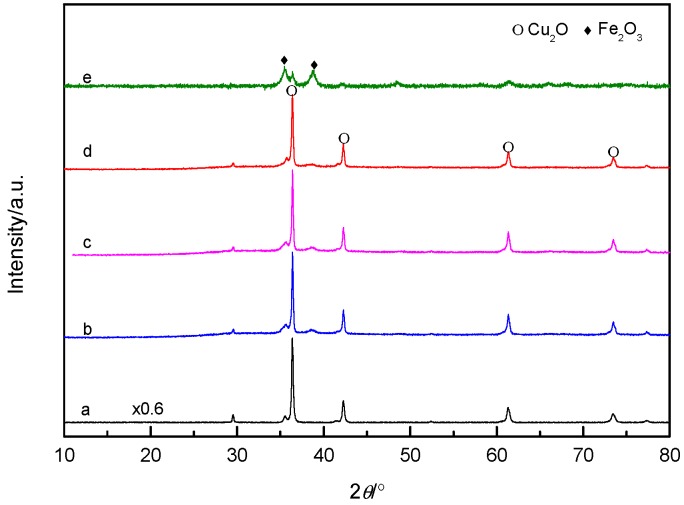
XRD patterns of Cu_2_O and Cu*_x_*O-Fe*_y_*O*_z_* with different Fe/Cu ratios (**a**) Cu_2_O, (**b**) CF-0.05, (**c**) CF-0.1, (**d**) CF-0.8, and (**e**) CF-2.

**Figure 2 nanomaterials-09-01301-f002:**
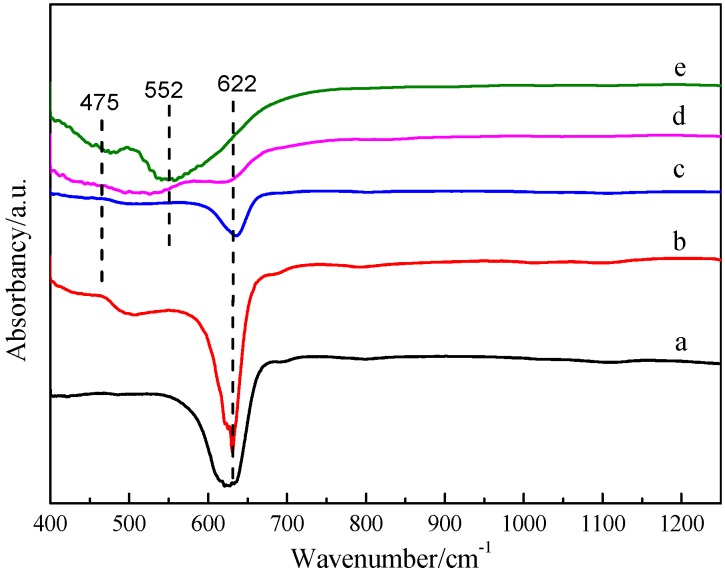
FT-IR spectra of Cu_2_O and Cu*_x_*O-Fe*_y_*O*_z_* with different Fe/Cu ratios (**a**) Cu_2_O, (**b**) CF-0.05, (**c**) CF-0.1, (**d**) CF-0.8, and (**e**) CF-2.

**Figure 3 nanomaterials-09-01301-f003:**
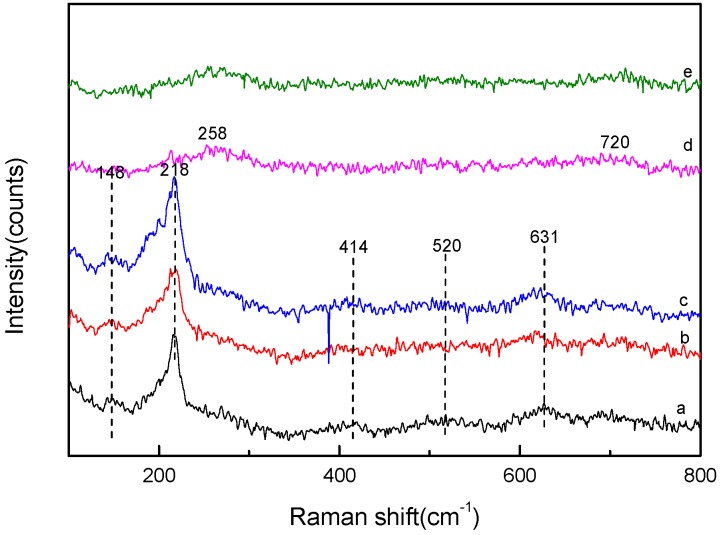
Raman spectra of Cu_2_O and Cu*_x_*O-Fe*_y_*O*_z_* with different Fe/Cu ratios (**a**) Cu_2_O, (**b**) CF-0.05, (**c**) CF-0.1, (**d**) CF-0.8, and (**e**) CF-2.

**Figure 4 nanomaterials-09-01301-f004:**
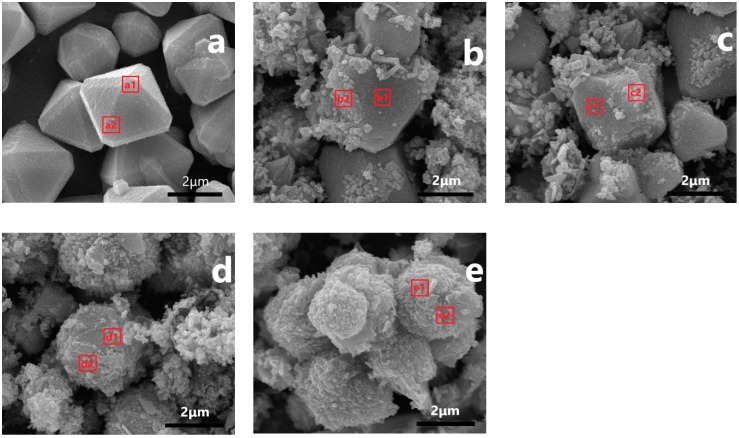
SEM images of Cu_2_O and Cu*_x_*O-Fe*_y_*O*_z_* with different Fe/Cu ratios (**a**) Cu_2_O, (**b**) CF-0.05, (**c**) CF-0.1, (**d**) CF-0.8, and (**e**) CF-2.

**Figure 5 nanomaterials-09-01301-f005:**
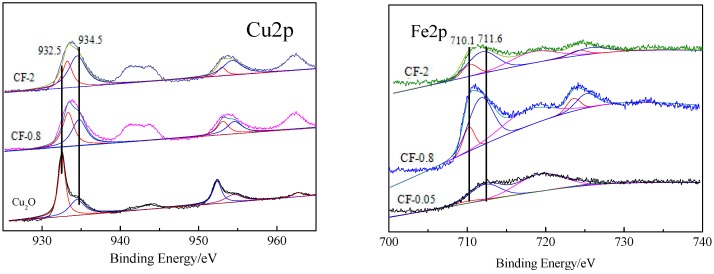
XPS spectra of Cu_2_O and Cu*_x_*O-Fe*_y_*O*_z_*_._

**Figure 6 nanomaterials-09-01301-f006:**
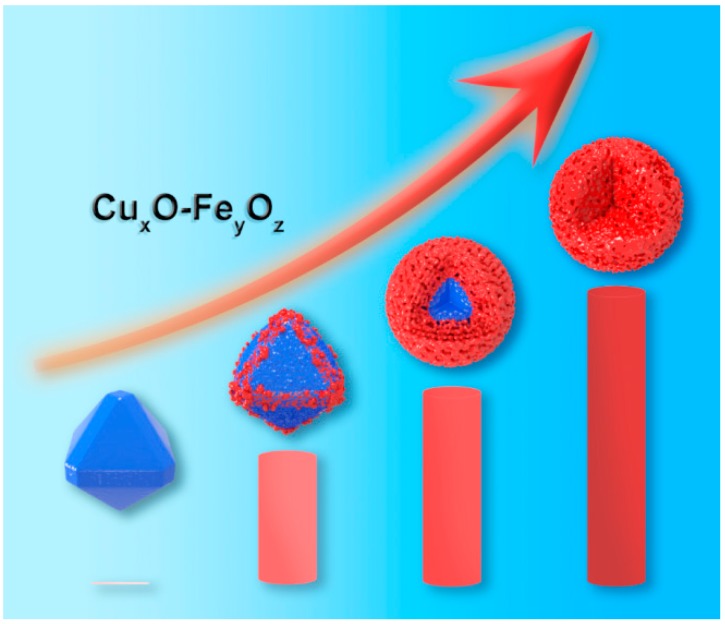
Formation of the Cu*_x_*O-Fe*_y_*O*_z_* structure.

**Figure 7 nanomaterials-09-01301-f007:**
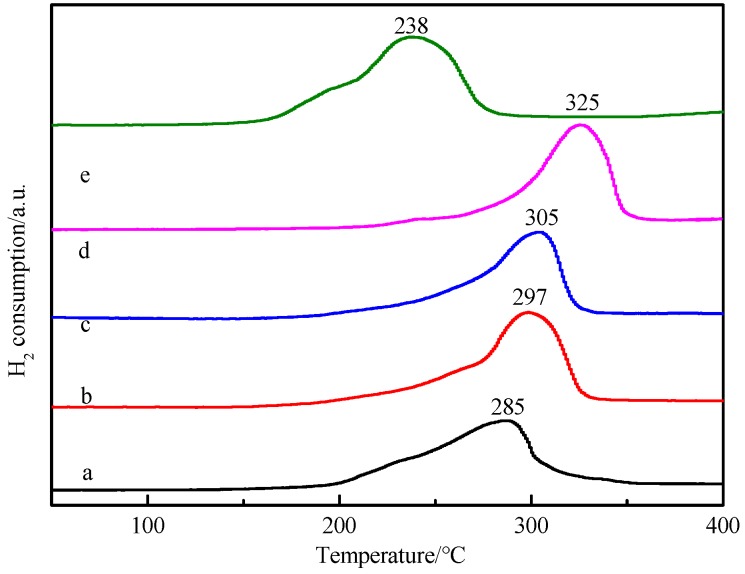
Temperature-programmed reduction (TPR) profiles of Cu_2_O and Cu*_x_*O-Fe*_y_*O*_z_* with different Fe/Cu ratios (**a**) Cu_2_O, (**b**) CF-0.05, (**c**) CF-0.1, (**d**) CF-0.8, and (**e**) CF-2.

**Figure 8 nanomaterials-09-01301-f008:**
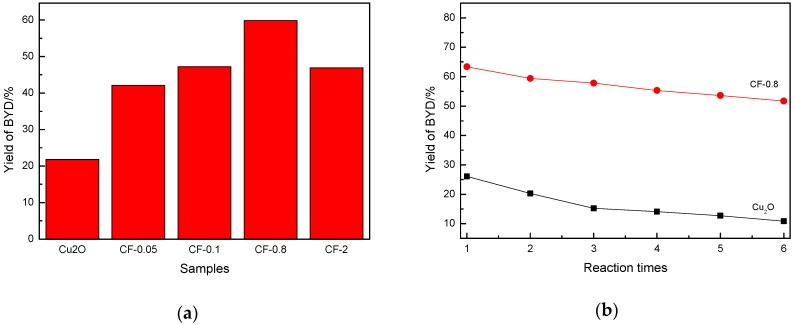
Catalytic performances of Cu_2_O and Cu*_x_*O-Fe*_y_*O*_z_*_._

**Table 1 nanomaterials-09-01301-t001:** EDS element analysis of different regions.

Catalysts	Region	AT%		
		OK	FeK	CuK
Cu_2_O	a1	7.67	0	92.33
	a2	8.97	0	91.03
CF-0.05	b1	7.85	0	92.15
	b2	13.91	10.46	75.63
CF-0.1	c1	8.67	0	91.33
	c2	17.36	14.84	67.80
CF-0.8	d1	29.33	35.74	34.92
	d2	29.12	35.97	34.91
CF-2	e1	33.23	41.96	24.81
	e2	34.12	41.72	24.16

**Table 2 nanomaterials-09-01301-t002:** Peak position and peak area ratio of different elements and valence states.

Catalysts	Fe(eV)		Fe^2+^/Fe^3+^ (Atomic)	Cu(eV)		Cu^2+^/Cu^+^ (Atomic)	Fe/Cu (Atomic)
	Fe^2+^2p_3/2_	Fe^3+^2p_3/2_		Cu^+^2p_3/2_	Cu^2+^2p_3/2_		
Cu_2_O	-	-	-	932.4	934.7	0.31	-
CF-0.05	710.1	711.6	0.07	-	-	-	0.26
CF-0.8	710.1	711.6	0.26	932.8	934.7	0.83	0.79
CF-2	710.1	711.6	0.32	932.8	934.7	2.3	1.23

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
