# Peer review of "Application of CuxO-FeyOz Nanocatalysts in Ethynylation of Formaldehyde"

_nanomaterials, 2019, doi:10.3390/nano9091301_

Round 1

Reviewer 1 Report

Composite nanomaterials have been widely used in catalysis. Among them, preparation of composite nanomaterials by redox method has attracted much attention. The nanomaterial contains Cu+, Cu2+, Fe3+, Fe2+. At certain ration core-shell structure is formed. The manuscript can be published after minor modification.

Is the title of 3.1 Section correct? In the text there is no Cu/SiO2 catalyst. The integrated area of the XPS peaks (especially at CF-0.8) are changing. Could be attributed this to the enrichment of Fe at the surface region? The area of Fe is higher at CF-08 than at CF-2. The interpretation of TPR deserves more analysis. The more comparison of TPR and XPs results would be useful. What is the driving fore of core-shell formation? Surface free energy?Please give some example from the literature (Langmuir, 26, 2010, 2167-2175, Catalysis Today 181, 2012, 163-170).

Author Response

Reply to reviewer #1:

The comments of reviewer #1:

Is the title of 3.1 Section correct? In the text there is no Cu/SiO2 catalyst.

Response: We thank the reviewer for pointing out this problem. We have changed the title of 3.1 "Structural analysis of CuO/SiO2 catalysts" to "Structural analysis of catalysts". The revised text was marked in red.

     2. The integrated area of the XPS peaks (especially at CF-0.8) are   changing. Could be attributed this to the enrichment of Fe at the surface region? The area of Fe is higher at CF-0.8 than at CF-2.

Response:

We thank the reviewer for the informative comments and instructive advice. In Figure S1, it is showed that as the Fe/Cu ratio is lower than 0.8, the peak area of Fe2p increases with the increase of Fe/Cu ratio, which is attributed to the enrichment of Fe on the surface of the catalyst particles. When the Fe/Cu ratio is further increased by 2, the peak area of Fe2p decreases. In response to this phenomenon, we analyze the O/Fe+Cu+O ratio on each sample, the results are listed in supporting information (Table S1). Table S1 shows that the ratio of O/Fe+Cu+O increases with the increase of Fe/Cu ratio. We speculate that the Fe2p integrated area of CF-2 decreases because of the increase of O content. The increase of O content in the surface is due to the conversion of Cu2O to CuO and, and the increase of O atoms bound to Fe3+. The above analysis and explanation have been added to line 263-271 and marked them in red.

    3. The interpretation of TPR deserves more analysis. The more comparison of TPR and XPs results would be useful.

Response:

According to the Suggestions of reviewers, we made a comparative analysis of XPS and H2-TPR data. “Combined with XPS analysis, the peak area of Fe2p increases with the increase of Fe/Cu ratio (Figure S1). When CF-0.8 reaches the maximum, it is showed that the enrichment of Fe on the surface of CF-0.8 reaches the maximum, a strong interaction between copper-iron complexes is formed, which makes the reduction temperature reach the highest. ” The above analysis and explanation have been added to line 339-343 and marked them in red.

What is the driving fore of core-shell formation? Surface free energy?Please give some example from the literature (Langmuir, 26, 2010, 2167-2175, Catalysis Today 181, 2012, 163-170).

Response:

We think the reviewer’s comments are very appropriate. As the reviewer has said, the driving force for nucleation is resulted from the difference in surface free energy[1-2]. Exposed high refractive index surfaces, defects, atomic steps, and kinks can be used as nucleation sites. The edge and vertex of Cu2O have higher surface free energy than those of the crystal mask, so it can be preferentially used as nucleation site[3]. The Fe3+s at these highly active nucleation sites are selectively deposited and grown to further form a core-shell structure.The above analysis and explanation have been added to line 308-311 and marked them in red.

Kiss, J.; Óvári, L.; Oszkó, A. ; Pótári, G.; Tóth, M.; Baán, K.; Erdóhelyi, A.Structure and reactivity of Au-Rh bimetallic clusters on titanate nanowires, nanotubes and TiO2(110).Catalysis Today. 2012, 181, 163-170. Óvári, L.; Berkó, A.; Balázs, N.; Majzik, Z.; Kiss, J. Formation of Rh-Au Core-Shell Nanoparticles on TiO2(110) Surface Studied by STM and LEIS. Langmuir, 2010, 26(3), 2167–2175. Zhu, H.; Du, M.L.; Wang, Y.; Wang, L.N.; Zou, M.L.; Zhang, M.; Fu, Y.Q. A new strategy for the surface-free-energy-distribution induced selective growth and controlled formation of Cu2O-Au hierarchical heterostructures with a series of morphological evolutions. J. Mater. Chem. A, 2013, 1, 919-929.

Reviewer 2 Report

This paper describes the synthesis of composite of CuxO-FeyOz, which were used as catalysts in ethynylation of formaldehyde. The following points should be considered.

As the authors mentioned, many catalysts have been reported for the reaction between formaldehyde and acetylene. The authors should compare these results with the newly established composites developed under the same reaction conditions in the manuscript. They also should emphasize the characteristic catalytic properties of the new catalysts in the reaction.

1,4-butynediol has isomers. The authors should specify the isomer. Propynol has the same problem. All the products structure must be characterized and confirmed.

In Figure 4, The SEM images show no scale bar. I cannot see the size of the clusters.

Round 2

Reviewer 2 Report

Ambiguity of the compound nomenclature must be clearly improved. For example, 1,4-butandiol means both 2-butyne-1,4-diol and 1-butyne-1,4-diol. We do not know which one is the correct product in this reaction.  The product characterization data are not available in the manuscript. The product structure must be characterized and confirmed. 

Propynol has also the same problem. In scientific paper, this kind of ambiguity must be  eliminated with evidence of analytical data.

Author Response

Reply to reviewer #2:

The comments of reviewer #2:

Ambiguity of the compound nomenclature must be clearly improved. For example, 1,4-butandiol means both 2-butyne-1,4-diol and 1-butyne-1,4-diol. We do not know which one is the correct product in this reaction.  The product characterization data are not available in the manuscript. The product structure must be characterized and confirmed.  Propynol has also the same problem. In scientific paper, this kind of ambiguity must be  eliminated with evidence of analytical data. Response:Thank you for the reviewer’s comments.The 1,4-butynediol described in this paper should be 2-butyne-1,4-diol. 2-butyne-1,4-diol is the only product for ethynylation of formaldehyde. In the previous work, we have confirmed by NMR and MS that there are no other forms of isomers. According to the reviewer's suggestion, we have identified the product. The 1,4-butynediol in the article has been changed to 2-butyne-1,4-diol, and the corresponding propyne alcohol has been changed to 2-propyn-1-ol. The modification part is marked red.

Round 3

Reviewer 2 Report

I can see the suitable corrections in the revised manuscript.